# Relationships between School Climate and Values: The Mediating Role of Attitudes towards Authority in Adolescents

**DOI:** 10.3390/ijerph19052726

**Published:** 2022-02-26

**Authors:** José Luis Gálvez-Nieto, Karina Polanco-Levicán, Ítalo Trizano-Hermosilla, Juan Carlos Beltrán-Véliz

**Affiliations:** 1Departamento de Trabajo Social, Universidad de La Frontera, Temuco 4780000, Chile; jose.galvez@ufrontera.cl; 2Departamento de Psicología, Universidad Católica de Temuco, Temuco 4780000, Chile; 3Departamento de Psicología, Universidad de La Frontera, Temuco 4780000, Chile; italo.trizano@ufrontera.cl; 4LICSA-Laboratorio de Investigación en Ciencias Sociales Aplicadas, Núcleo Científico Tecnológico en Ciencias Sociales y Humanidades, Universidad de La Frontera, Temuco 4780000, Chile; juan.beltran@ufrontera.cl

**Keywords:** school climate, values, attitudes towards authority, adolescence, positive development

## Abstract

School climate is related to a wide variety of positive results at the school level; however, its relationship with the construct of values has received little attention, despite being a key variable in the development of personality. This study aimed to examine the direct and indirect relationships between school climate, attitudes towards authority, and values. The participants in this study were 2683 students (51.2% men and 48.8% women) from 32 schools aged between 12 and 20 years (M = 15.78 years, SD = 1.35). Two models of structural equations were estimated, and the model that best fit the data confirmed that school climate was indirectly related to values through attitudes towards authority. The reciprocal and interactive relationships between school climate, attitudes towards authority, and values are also discussed.

## 1. Introduction

School climate is a highly relevant variable both to academic research and for society in general. There is a wide range of conceptualizations [1,2] regarding the definition of school climate; however, there is relative agreement that it is a complex construct with multiple dimensions [3].

School climate refers to the quality of social relationships and the character of school life [4,5,6], and has been used as a predictor or outcome variable in various empirical studies at the school level [1]. Specifically, the definition of school climate has to do with social and organizational aspects, including the relationships between school members and the values and norms shared by the educational institution [7]. One of the main socio-emotional factors that make up school climate is the teacher–student relationship. An interesting line of research suggests that students who perceive their teachers as understanding, respectful and willing to help them show a series of positive behaviors at school [8,9,10,11].

A positive school climate is a very important trait in the promotion of academic success, favoring not only academic achievement [12,13,14,15], but also prosocial behavior [16,17], the development of self-identity and self-esteem [18], delayed drug use [19], a reduced probability of assaults [20], and the prevention of behavioral problems [21]; it is also related to tolerance towards ethnic diversity [22,23]. In contrast, a negative school climate is associated with truancy [24], bullying [25,26], and school violence [27].

To understand the factors that configure school climate, various investigations have been based on Bronfenbrenner’s ecological systems theory [28]. This theory states that human development is a joint function of the person and the social context, emphasizing the interactive and reciprocal effects of the characteristics of the individual and the social subsystems that would directly and indirectly influence the development of individual behaviors. This theory is associated with the study of school climate, as individual behaviors are explained by the multiple school contexts that impact student development [1].

In the school context, the individual interacts with various variables located at different levels of the ecological system. The microsystem and mesosystem would be located at the proximal level. The school is located within the microsystem, and this subsystem is made up of direct interactions between members of the educational community, which gives character and tone to the school climate [1]. Sex is one of the individual variables that is relevant to school climate. For this reason, authors have carried out comparative studies between both groups. Some authors have evaluated the degree of factor equivalence between men and women [29,30]. Other authors have studied the differences between sexes [31,32], reporting that women present a better perception of school climate.

Thus, in schools with deteriorated relationships, various problems can arise both at the student level, through bullying [33], school violence [27], and substance use [19], as well as at the teacher level, through the victimization [34] and bullying of teachers [35]. When teacher–student social relationships are positive, they can significantly reduce the impact of the victimization produced by bullying among students [36] and promote socio-emotional development [18].

Continuing at the microsystem level, formal authority figures, such as teachers and normative references of the establishment, are highly relevant to adolescent development, since they contribute to adolescents having a better perception of the school climate [37]. The construct of attitude towards authority has been defined as the degree of importance attributed by adolescents to formally established authority figures, referring to school regulations, teachers, police, and laws [38]. Thus, respect for authority figures inside and outside of schools is related to favorable behavior in other social contexts where students participate [39], with positive relationships observed between teachers and students [37] and greater prevention of school violence [38]. On the other hand, students who present transgressive attitudes towards authority figures are more likely to practice cyberbullying [40], filioparental violence, problematic use of social networks, and alexithymia [41]. This attitude is also related to violence between parents and children and a deteriorating school climate [42].

The mesosystem is a function of the interactions between members of the educational community and very relevant variables are located in this subsystem. For example, positive relationships between family and teachers promote a better school climate, which subsequently results in a positive impact on student engagement [43], and both, in turn, are relevant factors that can explain academic performance [44].

Continuing at the mesosystemic level, a large proportion of the studies on school relationships in adolescence focus on multiple negative factors, for example, drug use, risky behavior, and bullying [14,19,45]. This focus on risk factors does not fully consider that adolescence is a stage in which it is possible to develop multiple potentialities. A theoretical approach that broadens this deficit-centered perspective is positive adolescent development, which emphasizes the existence of healthy conditions for optimizing adolescence [46].

It is in this context where values, understood from a positive adolescent development approach, become relevant. The value construct refers to cognitive representations of socially desired goals, which vary in importance and serve as guiding life principles that identify desired and appropriate behaviors in social interactions [47]. Specifically, this study follows the conceptual definition of the construct proposed by the adolescent positive development approach, in which values are a fundamental aspect of an adolescent’s personality (and would be composed of personal values, social values, and individualistic values [48]), since values regulate the behavior of the individual to promote their well-being, build healthier social relationships [49], develop a greater sensitivity to injustice, diversity and social inequality [50], and show a lower inclination towards materialism [51]. Therefore, the promotion of values could be a key element in developing good character in individuals, an aspect that could be related to a positive school climate and a means to reduce negative behaviors at school [52].

Values and their formation are influenced by different social subsystems [28]. At the microsystemic level, children learn values directly from their family, while at the mesosystemic level, the school stands out in the formation of students’ values [53]. At the exosystemic level, values are indirectly influenced by the neighborhood [37]. As for the macrosystem, studies highlight the indirect influences of culture, beliefs, and ideas of society [54].

Further from the individual is the exosystem, which is made up of a series of factors that must be considered as potential influences on school climate. For example, more distant environments such as the neighborhood or type of teaching significantly influence the individual behaviors of the students [37,55]. Regarding the type of teaching, evidence reports that scientific-humanistic schools presented a better perception of the school climate compared to technical-professional schools [30,37,56]. On the other hand, the macrosystem is made up of all those structures in which the student does not participate directly, such as values and culture. These factors influence the rest of the subsystems and the individual. For example, Sulak [55] carried out a study of 2560 schools, and showed that the establishments with the greatest problems with their school climate, bullying, and low respect for the authority of teachers were geographically located in high-crime areas. Finally, the chronosystem is the dimension of time, which includes the changes experienced by the student during their school trajectory and the social-historical moment in which the individual’s life takes place.

Considering the theoretical and conceptual background previously exposed, this study sought to fill the gaps in school research by examining the roles and relationships of the explanatory factors of the school climate, such as values and attitudes towards authority. While there are studies of great international relevance that highlight the relationship between values and school climate [3,4], that relationship has hardly been explored [52], which is why this study aims to examine the direct and indirect relationships between values, attitudes towards authority, and school climate. To reach this objective, this study proposed four hypotheses, which are represented based on Figure 1: (Hypothesis 1) values have direct and significant relationships with the school climate; (Hypothesis 2) values have direct and significant relationships with attitudes towards authority; (Hypothesis 3) attitudes towards authority have direct and significant relationships with school climate; and (Hypothesis 4) values have an indirect and significant relationship with school climate through attitudes towards authority. Additionally, sex and type of education were included as control variables in the model, given their potential effects [18,29,30,32,37,56].

## 2. Materials and Methods

### 2.1. Participants

The population was made up of adolescent students from public schools, subsidized private schools, and private schools. All students attended secondary education and lived in one of five macro-zones in Chile (*N* = 486,427). For the selection of the participants, a multistage, stratified probabilistic sampling procedure was used, with a reliability of 99.7%, a standard error of 2.48%, and a variance of p = q = 0.5 [57]. The sample considered the following strata: regions representative of the macro-zones of Chile; type of education and type of school. The first stage of the sample design consisted of the random selection of schools. Next, each school was asked for a current course list and, finally, the students were randomly selected.

The final sample was made up of (n) 2683 students from 32 schools (Table 1) of both sexes (51.2% men and 48.8% women) aged between 12 and 20 years (M = 15.78 years, SD = 1.35). The families of the students lived in urban (84.4%) and rural (15.6%) sectors. The selected schools included students from various socio-economic strata, but the majority represented low and medium socioeconomic levels.

### 2.2. Instruments

To answer the research objectives, four instruments were applied simultaneously. A sociodemographic questionnaire consisted of the following closed-ended questions: age, sex, grade, region, area, type of education (technical-professional or scientific-humanistic), type of school (public, subsidized private, private, or delegated administration corporation).

In addition, the Questionnaire to Assess School Climate (CECSCE) was applied. The CECSCE is a scale that evaluates school climate from the students’ perspective and has 14 items that are answered using a 5-point ordinal scale (from 1 = strongly disagree to 5 = strongly agree). Psychometric studies carried out in Chile provide evidence of their psychometric quality [58,59]. The CECSCE presents a two-dimensional factor structure. These factors are school climate (SC), which refers to peer relationships, security, and general feeling of well-being within the school (8 items, e.g., “My school is a very safe place”), and teacher climate (TC), which assesses academic demand, fairness, and treatment of students (6 items, e.g., “The teachers at this school are nice to students”). In this study, the CECSCE presented an adequate construct validity (WLSMV *χ*^2^ (df = 76) = 669,238, *p* < 0.001; CFI = 0.963; TLI = 0.955; RMSEA = 0.055). All factor loadings were statistically significant (*p* < 0.001). Reliability indices were also satisfactory: SC was linked to a Cronbach’s alpha of 0.780 (McDonald’s Omega = 0.785), and the TC factor was linked to a Cronbach’s alpha of 0.690 (McDonald’s Omega = 0.697).

A reduced version of the Scale of Values for Adolescent Positive Development (EVDPA-R) was also applied. The EVDPA-R assesses the perceived importance of a set of values based on the adolescent positive development approach. This scale has 12 items that are answered using a 7-point ordinal scale (from 1 = not important to 7 = most important). This scale was used for a psychometric study in Chile [48] that endorsed a structure of three factors. The first is called personal values (PV), and refers to the positive assessment of value aspects such as responsibility, moral integrity, and honesty (5 items, e.g., “Recognize and take responsibility when something has been done wrong”). The second, social values (SV), refers to the relationship and interaction of young people with different people in their community or neighborhood (5 items, e.g., “Belonging to or participating in social organizations”). The third is individualistic values (IV) and refers to the values of hedonism and social recognition, which can be considered as anti-values of an individualistic culture (3 items, e.g., “That other people admire me ”). In this study, the EVDPA-R presented an adequate construct validity (WLSMV *χ*^2^ (df = 51) = 2351.090, *p* < 0.001; CFI = 0.955; TLI = 0.942; RMSEA = 0.078). All factor loadings were statistically significant (*p* < 0.001). Reliability indices were also satisfactory: PV had a Cronbach’s alpha of 0.882 (McDonald’s Omega = 0.884), the SV factor was linked to a Cronbach’s alpha of 0.840 (McDonald’s Omega = 0.847), and IV was linked to a Cronbach’s alpha of 0.873 (McDonald’s Omega = 0.881).

Finally, an adapted version of the Attitudes to Institutional Authority in Adolescence Scale (AIA-A) was applied. The AIA-A is a self-report scale that assesses adolescent attitudes towards authority figures. The AIA-A has 9 items that are answered through a 5-point ordinal scale (from 1 = never to 5 = always). This scale was used in two psychometric studies in Chile [60,61]. The factorial structure of the AIA-A is made up of two factors: positive attitude to authority (5 items, e.g., “The police are there to make a better society for all”), referring to the degree of respect towards police and teachers; and positive attitude to transgression (4 items e.g., “It is normal to break the law if no one is harmed”), referring to positive attitudes towards rule transgression.

In this study, the AIA-A presented an adequate construct validity and the confirmatory factor analysis presented adequate goodness-of-fit indices (WLSMV *χ*^2^ (df = 26) = 453.503, *p* < 0.001; CFI = 0.968; TLI = 0.955; RMSEA = 0.074). All factor loadings were statistically significant (*p* < 0.001). AIA-A factors presented adequate reliability indices: positive attitude to authority was linked to a Cronbach’s alpha of 0.716 (McDonald’s omega = 0.738), and positive attitude to transgression was linked to a Cronbach’s alpha of 0.775 (McDonald’s omega = 0.786).

### 2.3. Procedures

For the application of the measurement instruments, the school directors were contacted and asked to sign an agreement to carry out the study. Subsequently, informed consent forms were sent to the parents or guardians of the students. Once the authorizations were obtained, the students submitted an informed consent form. The ethical safeguards of this project were endorsed by the ethics committee of the Universidad de La Frontera de Chile.

### 2.4. Data Analysis

Medians, standard deviations, and polychoric correlations between the latent variables were estimated. Furthermore, a confirmatory factor analysis (CFA) and structural equation models (SEM) were evaluated using the MPLUS v.8.1 software (Muthén & Muthén, Los Angeles, CA, USA) [62]. Structural equation models (SEMs) are a family of multivariable statistical models that provide greater flexibility than regression models; for instance, they allow the inclusion of some measuring errors in dependent and independent variables. SEMs have two basic com-ponents: (1) the structural model, which is a path model allowing the relation between dependent and independent variables to be determined; and (2) the measurement model, which allows the researcher to use a set of variables (indicators) to measure one (latent) variable, either dependent or independent [63]. In order to adequately select the measurement model, a multivariate normality test was conducted (kurtosis test = 6.935; *p* < 0.001). Results were statistically significant, so, given the absence of multivariate normality, a robust estimator was selected. All models used suitable methods for the analysis of categorical variables, such as a matrix of polychoric correlation matrix and the weighted least squares means and variance adjusted (WLSMV) estimation method. The following goodness-of-fit indexes were used to evaluate the CFA models: WLSMV-*χ*^2^, comparative fit index (CFI), Tucker–Lewis index (TLI), and root mean square error of approximation (RMSEA). For CFI and TLI, values equal to or greater than 90 were considered a reasonable fit [64]. For RMSEA, values less than or equal to 0.080 were considered a reasonable fit [65]. The variables sex (0 = male; 1 = female) and type of teaching (0 = technical-professional; 1 = scientific-humanistic) controlled the relationships of the mediating and outcome variables. For the reliability estimation, McDonald’s ω and Cronbach’s α coefficients were used [66,67].

## 3. Results

Table 2 shows the medians, minimums, maximums, and paths between the latent variables. Statistically significant correlations were observed between all the variables studied. In general, positive correlations were observed between the dimension of favorable attitude towards authority and school climate, while negative correlations were observed between positive attitude towards transgression and teacher climate. The factors of the scale of values presented significant correlations with the factors of the scale of attitude towards authority and school climate.

### 3.1. Complete Structural Model

Figure 2 shows the complete structural model, which included correlations between all the variables. Goodness of fit indices were satisfactory (WLSMV *χ*^2^ (df = 602) = 5243.543, *p* < 0.001; CFI = 0.941; TLI = 0.935; RMSEA = 0.055 (CI 90% = 0.053–0.056)). As can be seen, the factors personal values and social values did not show direct relationships with school climate. The individualistic values factor presented significant, but very weak relationships with both factors of school climate (β = 0.068, *p* < 0.001; β = 0.096, *p* < 0.001). Subsequently, in order to improve the fit of the model, non-significant or low-association pathways were eliminated. Under these criteria, the direct relationships between values and school climate were eliminated.

### 3.2. Structural Model with Mediated Relationships

The structural model of mediated relationships presented satisfactory goodness of fit indices (WLSMV *χ*^2^ (df = 608) = 5186.944, *p* < 0.001; CFI = 0.942; TLI = 0.937; RMSEA = 0.054 (CI 90% = 0.053–0.056)). Figure 3 shows the final model with the standardized coefficients. In this figure, the relationships with both direct and indirect influence on the school climate are presented. The model shows indirect and statistically significant relationships between values and school climate, which are mediated by attitudes towards authority.

The direct relationship showed that personal values and social values were negatively related to a positive attitude towards transgression (β = −0.206, *p* < 0.001; β = −0.175, *p* < 0.001, respectively) and positively related to favorable attitude towards authority (β = 0.207, *p* < 0.001; β = 0.180, *p* < 0.001, respectively). Consistently, the individualistic values presented negative relationships with positive attitude towards authority (β = −0.139, *p* < 0.001) and positive relationships with favorable attitude towards transgression (β = 0.194, *p* < 0.001).

The direct relationship between attitudes towards authority and school climate showed that the factor positive attitude towards transgression presented significant and negative relationships to both teacher climate and school climate (β = −0.186, *p* < 0.001; β = −0.161, *p* < 0.001, respectively). Finally, favorable attitude towards authority presented significant and positive relationships with teacher climate and school climate (β = 0.886, *p* < 0.001; β = 0.695, *p* < 0.001, respectively). Consequently, the model that best fits the data is the model of mediated relationships (Figure 3).

### 3.3. Control Variable

As can be seen (Figure 2 and Figure 3), the control variable sex (1 = woman) presented a statistically significant and negative relationship with positive attitude towards transgression (β = −0.148, *p* < 0.001; β = −0.203, *p* < 0.001, respectively). In the case of the teaching type variable (1 = scientific-humanistic), it presented a statistically significant and positive relationship with school climate (β = 0.102, *p* < 0.001; β = 0.184, *p* < 0.001, respectively).

## 4. Discussion

This study aimed to examine the direct and indirect relationships between values, attitudes towards authority, and school climate. The findings of this investigation supported three of the four hypotheses. In general terms, the model that best fits the data confirms that values are indirectly related to school climate through attitude towards authority.

The first hypothesis posed the following: (h1) values have direct and significant relationships with school climate. The findings of this study allowed us to reject h1, since the dimensions involving the construct “values” (personal values and social values) did not present a statistically significant path with both factors of school climate. The rejection of the hypothesis of direct relationships between values and school climate can be explained in this way: values, despite being a construct that is formed from multiple interactions with the environment, is nevertheless an individual measure that evaluates attitudes developed at the personal level [48,50]. On the other hand, school climate constitutes a group phenomenon [7], which extends beyond individual experience [1,2,4], is based on school experience schemes, and is shaped by the lived experience of social and organizational interactions with the school [5,6].

The second hypothesis posed the following: (h2) values have direct and significant relationships with attitude towards authority. The results allowed us to maintain this hypothesis, as social values and personal values were found to have significant and negative relationships with attitudes towards transgression and positive relationships with favorable attitudes towards authority; in the same way, the individualistic values factor presented positive relationships with the factor attitudes towards transgression and negative relationships with the factor favorable attitude to authority. In this regard, personal values, which characterize adolescents with higher levels of responsibility, moral integrity, and honesty, are positively related to attitudes of respect for authority, which refers to school regulations, teachers, and the police [38]. These results are consistent with research that suggests that students with favorable attitudes towards authority show greater respect for their teachers [37]. The results regarding the positive relationship between individualistic values and transgressive attitudes are consistent with previous studies, given that students who show a greater tendency towards individualistic values (hedonism) present a higher cannabis consumption [68], greater tendency towards materialism [51], and deteriorated school climate [42].

Regarding the third hypothesis—(h3) attitudes towards authority have direct and significant relationships with school climate—the results allowed us to maintain this hypothesis, given that the factor attitude towards transgression presented negative relationships with both factors of school climate. On the other hand, the factor favorable attitude towards authority presented significant and positive relationships with both factors of school climate. In this regard, research in the field suggests that students who show high levels of rule transgression also show high levels of violence towards their parents and low levels of school climate, an aspect that would directly impact their social relationships at school [42]. The results of this study are consistent with research that suggests that students who perceive healthy school climates and respectful relationships with their teachers tend to have low levels of transgression towards authority [37,60].

Considering these antecedents, a positive attitude towards the rules and institutions such as the school and police not only protects children against participation in violent behavior, but also favors academic success [15] and psychosocial adjustment in other social contexts [38].

Finally, the fourth hypothesis posed the following: (h4) values have an indirect and significant relationship with school climate through attitudes towards authority. The results suggest maintaining this hypothesis, given that the model with the best fit was the model of relationships between school climate and values mediated by attitudes towards authority. This study highlights that school climate, which is located in the microsystem [1], is indirectly influenced by values in personal, social, and individualistic dimensions through attitudes towards authority. It is relevant to note that these values are primarily obtained in the family microsystem [53], but are also influenced by more distant subsystems, such as the macrosystem [54], impacting individual and group behaviors such as school climate [7]. In this sense, it is relevant to point out that the promotion of adolescent values, which can be learned in the family, educational institutions, or society, will indirectly and positively impact school relationships, which at the same time will be influenced by attitudes towards authority referents. This triad is relevant in the context of adolescence, which, as indicated in the adolescent positive development approach, is an evolutionary stage of great plasticity in which adolescents can achieve healthy development by displaying all of their potentialities [46]. It is relevant to note that the effects between the variables were controlled according to sex and type of teaching, and, in this regard, both variables were statistically significant in the model. The significant effect of the sex variable on positive attitude to transgression stands out, where it can be observed that men presented higher levels of transgression. These results are consistent with previous literature [37,39,40]. Regarding the effects of type of teaching, the results presented a statistically significant relationship with school climate, where it can be observed that the scientific-humanistic schools presented a better perception of school climate, compared to the technical-professional schools. These results are consistent with previous studies [30,37,56].

In relation to the theoretical implications of the study, this research contributes to the theoretical development of educational psychology by examining factors that promote or hinder school climate. The interpretive framework that allowed these relationships to be understood is Bronfenbrenner’s theory of ecological systems [28]. In this regard, it is necessary to highlight that individual behaviors are explained by different variables located in different subsystems, each one nested within the other.

The formation of school climate is complex and can be the result of multiple influences at the proximal and distal levels of the educational system. The characteristics of students, teachers, and staff can be considered as factors related to the internal development of school climate [11,69].

Another relevant theoretical contribution of the present study has to do with the evaluation of values using the positive adolescent development approach [48]. This approach focuses on the implementation of educational interventions aimed at promoting healthy behaviors [46], rather than risk factors, which are the focus of the approach which predominates in studies in the school setting [14,19,45].

Since school climate is a factor that can be modified [2,55], the work of professionals in the educational field could be oriented to modify and eventually improve perceptions of the school climate, altering this construct in an indirect way by modifying values. Taking into account student perceptions of the school climate can help guide the selection and implementation of school improvement efforts, and would also make students more receptive to implemented actions [23]. In this sense, the findings of this study suggest that, by stimulating social and personal values and, conversely, decreasing the trend towards the development of individualistic values, it is possible to strengthen teachers’ sense of connection to their schools and social relationships between peers. This in turn could improve the students’ school experience, resulting in academic success [14,15,17,22]. Thus, professional teams linked to promoting schools’ climates must intervene first in adolescent values in relation to developing respect for teachers and authority figures in the school, so that, in an indirect way, the levels of school climate will significantly improve in educational institutions.

Schools must assume the permanent role of teaching and cultivating values, an approach which has been called education in values [70]. This approach underscores the value of communicating the desired values to students during their time in school as part of their comprehensive training [71]. Therefore, due importance should be given to the curriculum content related to values [72].

It is important to note that the intervention work of schools must be rooted in the ecological model [28] and focused on the potential of adolescents, expanding past the perspective focused on risk factors [46] to include the family, given its importance in the formation and transmission of values [53]. In addition, interventions should consider factors located in more distant ecological subsystems that indirectly affect student interactions, such as the assets of the neighborhood [37,55].

Despite the indirect influence of distal and structural factors located at the exosystemic and macrosystemic level [28], the school plays a fundamental role in the socialization processes of adolescent students, and can be effective in the intervention of the psychosocial environment and teaching of values [52].

The results of the present study should be interpreted with caution, because, like most studies of school climate, they follow a cross-sectional design. This is a problem since it is not possible to reflect the dynamic and changing nature of the school climate over time [2,18,73]. It would be beneficial if future lines of research carried out studies using more robust methodologies, such as longitudinal designs. Furthermore, this study was based on self-report measures provided by adolescent students. Although the convenience of studying school climate from the perspective of students has been documented, recent studies have shown the need to measure school climate from multiple perspectives [2,3,74].

## 5. Conclusions

The present study has provided an explanatory model of the direct and indirect relationships between values, attitudes towards authority, and school climate. The structural relationship model that best fits the data established that values were indirectly related to school climate through attitudes towards authority, controlling for the effect of sex and type of teaching. This model was necessary to explain the importance of values in the social relations of adolescents, and it provides a relevant interpretive framework for the creation of future intervention programs in the field of educational psychology. When it comes to strengthening school climate in the adolescent stage, these findings generally show the importance of strengthening social values, that is, stimulating participation in social organizations, dedicating time to helping others, and personal values aspects such as honesty and responsibility. One aspect to pursue is decreasing the tendency towards individualistic values, which are those in which adolescents show a tendency towards hedonism, or a tendency towards transgressive negative relationships with their peers and teachers. These factors would be essential components of efforts to indirectly and significantly stimulate healthy relationships in schools.

## Figures and Tables

**Figure 1 ijerph-19-02726-f001:**
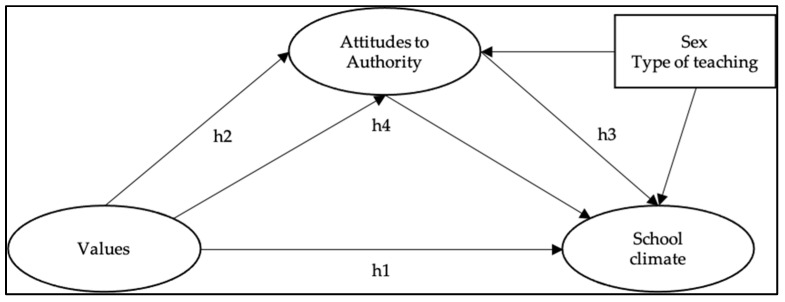
Theoretical model of the structural relationships between values, attitude towards authority, and school climate. Note: h = hypothesis.

**Figure 2 ijerph-19-02726-f002:**
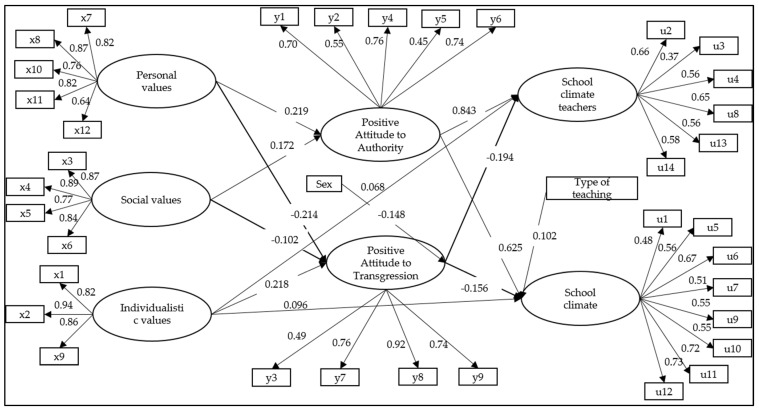
Complete model of structural relationships between values, attitudes towards authority and school climate.

**Figure 3 ijerph-19-02726-f003:**
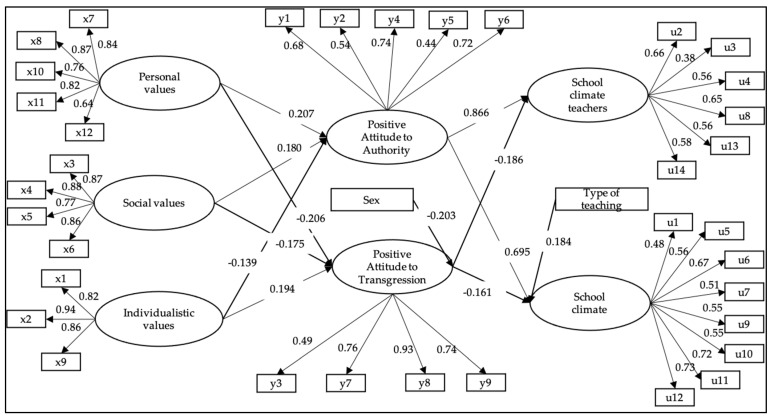
Model of structural relationships between school climate and values through attitude towards authority. Note: standardized coefficients are presented; non-significant paths were not included.

**Table 1 ijerph-19-02726-t001:** Sample demographics.

Variable	*n*	%	% Women	% Men
Type of school				
Public	801	29.86	14.65	15.21
Subsidized private	1523	56.76	27.80	28.96
Private	257	9.58	4.32	5.26
Delegated administration corporation (Law 3166)	102	3.80	1.98	1.83
Total	2683	100	48.8	51.2
Region				
Region of Antofagasta	192	7.16	3.47	3.69
Region of Coquimbo	83	3.09	1.49	1.60
Metropolitan Region	1643	61.24	30.64	30.60
Region of La Araucanía	695	25.90	12.19	13.72
Region of Magallanes and Chilean Antarctica	70	2.61	0.97	1.64
Total	2683	100	48.8	51.2
Type of teaching				
Scientific-humanistic	1695	63.18	35.22	27.95
Technical-professional	988	36.82	13.53	23.29
Total	2683	100	48.8	51.2

**Table 2 ijerph-19-02726-t002:** Covariance matrix (polychoric correlation), descriptive statistics.

	F1 (IV)	F2 (SV)	F3 (PV)	F4 (PAA)	F5 (PAT)	F6 (SC)	F7 (TC)
F1. Individualistic values (IV)	1						
F2. Social values (SV)	0.517 **	1					
F3. Personal values (PV)	0.307 **	0.474 **	1				
F4. Positive attitude towards authority (PAA)	0.188 **	0.255 **	0.285 **	1			
F5. Positive attitude towards transgression (PAT)	0.077 **	−0.059 **	−0.144 **	−0.172 **	1		
F6. School climate (SC)	0.187 **	0.251 **	0.268 **	0.529 **	−0.130 **	1	
F7. Teacher climate (TC)	0.180 **	0.248 **	0.271 **	0.616 **	−0.165 **	0.626 **	1
Median	−0.1668	−0.0005	0.2653	−0.0650	−0.1528	0.0814	0.0983
Minimum	−1.64	−2.04	−3.90	−3.24	−1.48	−3.56	−3.12
Maximum	2.15	2.04	1.10	2.05	2.77	2.38	2.33

Note: the variables were standardized. ** *p* < 0.001 (bilateral).

## Data Availability

No appliable.

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
