# Peer review of "Relationships between School Climate and Values: The Mediating Role of Attitudes towards Authority in Adolescents"

_ijerph, 2022, doi:10.3390/ijerph19052726_

Round 1

Reviewer 1 Report

It´s a very interesting research, but the factors are studied a lot of years ago. Coul be more interested to add the influences of ITC in their values, attitudes and respect for authority in a next study. So, could be interested to know if there differences between privates and concertated centres.

Good look 

Author Response

We thank the revisor’s comments. These aspects will be included in future studies.

Reviewer 2 Report

The introduction provides sufficient background and relevant references. Please, review the following part:

  • Line 53. In the text says "The microsystem and exosystem would be located at the proximal level". Please, review if there is a mistake and it shoud say "mesosystem" where it says "exosystem".

The design is appropiate. Please, review the following part:

  • Line 229/230 (page 6). In the text says "The variables ... and type of school (0 = Technical-professional; 1 = Scientific-humanistic) ..." Nevertheless, in page 4 (both in table 1 and in the description of the sociodemographic questionnaire) says that there are three type of school (public, subsidized private and private); and two type of teaching/education (scientific-humanistic, and technical-professional). Please, review if in page 6 it it could say "type of teaching" where it says "type of school".

The part of statistics is very good, congratulations. Only two things:

  • The authors should clarify that their statistical assumptions have been revised for the application of SEM. This would provide clarity and aid future replication. E.g., as they employ WLSMV normality migh not be assumed, but this should be better explained.
  • Line 243"included correlations between all the variables" Are authors sure that this is referred to correlation or weights of the model?

Author Response

Point 1: The introduction provides sufficient background and relevant references. Please, review the following part: Line 53. In the text says "The microsystem and exosystem would be located at the proximal level". Please, review if there is a mistake and it shoud say "mesosystem" where it says "exosystem".

Response 1: Thank you, we have corrected the mistake.

Point 2: Line 229/230 (page 6). In the text says "The variables ... and type of school (0 = Technical-professional; 1 = Scientific-humanistic) ..." Nevertheless, in page 4 (both in table 1 and in the description of the sociodemographic questionnaire) says that there are three type of school (public, subsidized private and private); and two type of teaching/education (scientific-humanistic, and technical-professional). Please, review if in page 6 it it could say "type of teaching" where it says "type of school".

Response 2: Thank you, we have corrected the mistake.

Point 3: The authors should clarify that their statistical assumptions have been revised for the application of SEM. This would provide clarity and aid future replication. E.g., as they employ WLSMV normality migh not be assumed, but this should be better explained.

Response 3: The revision of the statistical assumptions for the application of SEM has been included, normality and multivariate normality (Mardia-test).

Point 4: Line 243"included correlations between all the variables" Are authors sure that this is referred to correlation or weights of the model?

Response 4: We have exchanged the concept of “correlations” for “paths”, which is more appropriate for SEM.

Reviewer 3 Report

This research explores an interesting adolescents field. It is  not very  well known. The research is very difficult because it is measuring individual and social factors. The methodology is adequate.

I would recommend  the authors not to investigate so many questions since it would be necessary to use more techniques to deepen the results or explain more the tecniques used to get the results. These are scarce. Also the conclusions.

Author Response

Point 1: This research explores an interesting adolescents field. It is  not very  well known. The research is very difficult because it is measuring individual and social factors. The methodology is adequate.

Response 1: We appreciate your comments for they have helped us improve the manuscript.

Point 2: I would recommend  the authors not to investigate so many questions since it would be necessary to use more techniques to deepen the results or explain more the tecniques used to get the results. These are scarce. Also the conclusions.

Response 2: In “Data Analysis” we have clarified the SEM techniques, thus improving the understanding of our conclusions.

Reviewer 4 Report

The aims are interesting, innovative and of high educational relevance.
The bibliographical references are current, relevant, with an appropriate balance between those from the national sphere in which the study was carried out (Chile) and those from the international sphere.

The methodology is very appropriate for achieving the planned aims, using methodological rigour and with a large sample of subjects that allows us to assess its potential representativeness for the population as a whole. The methodological design, samples, data collection procedure and data analysis are presented in a detailed and appropriate manner. The fit values of the causal/structural model are also adequate.

I consider that there is a good starting approach that provides sufficient justification for the study and also an adequate treatment and analysis of the data. Some relatively low values in the Cronbach's Alpha reliability (below 0.80) or the absence of more specific analyses of reliability and convergent validity of the scales (such as the Composite Reliability tests or AverageVariance Extracted (AVE)) could perhaps be pointed out.

The conclusions and approaches presented reflect maturity and creativity, relating the results to the theoretical framework.

Moreover, the contributions are transferable to the educational context.

Author Response

Point 1: The aims are interesting, innovative and of high educational relevance.

The bibliographical references are current, relevant, with an appropriate balance between those from the national sphere in which the study was carried out (Chile) and those from the international sphere.

The methodology is very appropriate for achieving the planned aims, using methodological rigour and with a large sample of subjects that allows us to assess its potential representativeness for the population as a whole. The methodological design, samples, data collection procedure and data analysis are presented in a detailed and appropriate manner. The fit values of the causal/structural model are also adequate.

Response 1: We appreciate your comments.

Point 2: I consider that there is a good starting approach that provides sufficient justification for the study and also an adequate treatment and analysis of the data. Some relatively low values in the Cronbach's Alpha reliability (below 0.80) or the absence of more specific analyses of reliability and convergent validity of the scales (such as the Composite Reliability tests or AverageVariance Extracted (AVE)) could perhaps be pointed out.

Response 2: Some of the reliability factors are indeed relatively low (alfa <0.8). Nonetheless, the scales had previous psychometric studied backing up their use in Spanish and in the Chilean context. More specific psychometric parameters were not estimated (complex reliability and mean variance). However, Cronbach alfa coefficient, McDonald’s omega, and confirmatory factorial analysis were used as reliability indicators and measure model, respectively.

Point 3: The conclusions and approaches presented reflect maturity and creativity, relating the results to the theoretical framework. Moreover, the contributions are transferable to the educational context.

Response 3: We appreciate your comments.